

**Sensitivity to species selection indicates the effect of nuisance variables on marine microfossil transfer functions**

Lukas Jonkers and Michal Kučera

MARUM – centre for marine environmental research, Bremen University, Leobener Straße 8, 28359 Bremen, Germany.

Keywords: planktonic foraminifera, transfer functions, MARGO, WA, MAT

**Abstract**

The species composition of many groups of marine plankton appears well predicted by sea surface temperature (SST). Consequently, fossil plankton assemblages have been widely used to reconstruct past SST. Most applications of this approach make use of the highest possible taxonomic resolution. However, not all species are sensitive to temperature and their distribution may be governed by

other parameters. There are thus reasons to question the merit of including information all species, both for transfer function performance and for its effect on reconstructions.
Here we investigate the effect of species selection on planktonic foraminifera transfer functions. We assess species importance for transfer function models using a random forest technique and evaluate the performance of models with increasing number of species. Irrespective of using models

that use the entire training set (weighted averaging) or models that use only a subset of the training set (modern analogue technique), we find that the majority of foraminifera species does not carry useful information for temperature reconstruction. Less than one third of the species in the training set is required to provide a temperature estimate with a prediction error comparable to a transfer function that uses all species in the training set. However, species selection matters for

paleotemperature estimates. We find that transfer function models with different number of species but with the same error may yield different reconstructions of sea surface temperature when applied on the same fossil assemblages. This ambiguity in the reconstructions implies that fossil assemblage change reflects a combination of temperature and other environmental factors. The contribution of the additional factors is site and time specific, indicating ecological and geological complexity in the

formation of the sedimentary assemblages. The possibility of obtaining multiple different reconstructions from a single sediment record presents a previously unrecognised source of uncertainty for sea surface temperature estimates based on planktonic foraminifera assemblages. This uncertainty can be evaluated by determining the sensitivity of the reconstructions to species pruning.




**1 Introduction**

Any method to infer quantitative paleoenvironmental information from (fossil) species assemblages
relies on the use of a modern (or near modern) training set to identify a statistical relationship
between the species assemblage and the environmental variable to be reconstructed. A range of

methods exists, varying broadly in the way the training set is used to define the relation between the
species community and the environmental parameter of interest. In this context, one aspect of the
training set design that has received little formal attention, is species selection. In applications based
on marine microfossils, the assumption has been that (nearly) all species potentially carry useful
information and training sets have been designed to include as many species as practically possible

(Kucera et al., 2005). However, there are fundamental, theoretical reasons to question if full
taxonomic resolution is necessary for accurate reconstructions. Firstly, species may respond to other
environmental variables than the one of interest and therefore confound the reconstruction. Such
species would negatively impact the predictive power of the transfer function model. Secondly,
species could be opportunistic or have a wide ecological niche and hence provide little information

on the environmental variable to be reconstructed. These species would add no, or very little,
information to constrain the reconstruction. The premise of full taxonomic resolution has been
recently tested by Juggins et al. (2015), who demonstrated that transfer function techniques are
variably sensitive to the influence of non-informative taxa. These authors used highly diverse
assemblages from coastal and lacustrine environments, where the proportion of uninformative or

confounding taxa can be expected to be high and where species selection for transfer function
models is intuitively appropriate.

However, marine microfossil assemblages (notably planktonic foraminifera) are less diverse and the
species display a low degree of endemicity. In these circumstances, it would appear important to
retain as many species in the training set as possible. Indeed, up to now, species pruning in marine

microplankton applications has been done, if at all, for practical, rather than ecological reasons.
Species were removed or lumped because of low abundance, preservation or taxonomic issues
(Hayes et al., 2005; Vernal et al., 2001; Zielinski et al., 1998). However, the minimum number of taxa
required for a robust transfer function model, or the influence of non-informative or nuisance taxa
on such models, has never been formally evaluated. This is remarkable, considering that many

species of marine microplankton appear to have similar response to the modelled variables and that
there is also evidence for responses to multiple variables (Telford et al., 2013; Siccha et al., 2009;
Steinke et al., 2008). Furthermore, from a practical point of view, reducing the number of species in
the model may improve counting statistics and reduce counting time. Moreover, reducing taxonomic
resolution may also be beneficial for application of automated identification and counting systems

(e.g. Beaufort and Dollfus, 2004; Hsiang et al., 2016) and enable the use of legacy datasets with



incomplete taxonomy. The lack of a formal evaluation of species pruning in marine microfossil studies is also relevant because of the prolific use of different transfer function techniques. The evaluation of species pruning by Juggins et al. (2015) was carried out for 'global models' where the entire training set is used to define the species response curve to an environmental variable. Such

methods are used on marine microfossil applications as well (e.g. Imbrie and Kipp method (Imbrie and Kipp, 1971), weighted averaging (WA) or artificial neural networks (Malmgren and Nordlund, 1997)). However, in the marine studies the most popular approach is the modern analogue technique (MAT). This approach uses similarity measures to identify a small 'local' subset of samples from the training set to derive the environmental variable to be reconstructed. Such a 'local'

approach is fundamentally different and could be sensitive to species pruning in the training set in a different way.

Here we evaluate species importance for transfer function performance for two widely differing methods (WA and MAT), representing both ends of the spectrum between local and global methods. We start by assessing species importance to determine the ranking of species for transfer function

performance. We then investigate why some species are more important than others. Finally, because the behaviour of transfer functions models cannot be assessed within the modern training set only, we assess the influence of species selection on paleotemperature reconstructions.

We use planktonic foraminifera as a model group, but the results from this study are likely of wider relevance to paleoecological reconstructions based on marine microfossils. Planktonic foraminifera

are ideal for this purpose because of the existence of a large global training set (Siccha and Kucera, 2017) and because of strong evidence that their distribution in surface sediments can be predicted by sea surface temperature alone (Bé and Hutson, 1977; Morey et al., 2005).

## 2 Data and Methods

To derive the transfer functions, we use planktonic foraminifera assemblage data from core top sediments recently compiled in the ForCenS dataset (Siccha and Kucera, 2017). Multi-species categories and morphotypes were removed and in case of replicate samples (at the same location) one sample was randomly selected. Annual mean temperature was assigned to each sample in the training set based on data from climatology (Stephens et al., 2002) averaged over the upper 50 m

and within a 100 km radius of each core top sample. To test the effect of species selection on the performance of the transfer function outside of the calibration dataset, we analysed three fossil datasets. To evaluate the effect on assemblages with different diversity, we use two downcore records from the Atlantic: M35003-4 from the Caribbean (Hüls and Zahn, 2000), which spans 0-55 ka BP, and a longer, 0-180 ka BP, record from the Iberian Margin: MD95-2040 (De Abreu et al., 2003).

And to evaluate the effect on past spatial SST patterns, we reanalyse the MARGO Last Glacial



Maximum (LGM) dataset from the North Atlantic Ocean (Kucera et al., 2005). The taxonomy of all fossil samples was harmonised with the ForCenS dataset following the same criteria as in Siccha and Kucera (Siccha and Kucera, 2017). Species not reported to be present in the downcore assemblages were assumed to be absent.

To determine how many and which species are needed for the transfer function models we start with assessing species importance following the approach described by Juggins et al. (2015). Briefly, this involves randomly selecting 1/3 of the species in the training set and dividing this reduced training set in a part that is used to build the transfer function model and an out-of-bag (OOB) part that is used to evaluate the model. The proportion of each species in the OOB selection is then

randomly permuted and the performance of the model is reassessed. Species for which the permutation leads to an increase in the prediction error are considered important. The species importance is derived from the average difference in prediction between the OOB and the permuted OOB samples across 1,000 bootstrap samples of the training set. The entire approach is repeated 10 times in order to get an error estimate on the species importance. For WA we use inverse

deshrinking to obtain the environmental variables and for MAT we use the 5 closest analogues determined using squared chord distance. We note that the way the species importance is assessed and the transfer function models with different numbers are designed is implicitly using a rest group of non-included species. This is because the calculations are based on the relative abundances of the species in the training set with the full taxonomic resolution. This procedure simulates a situation

where the total number of fossils of the studied group has been determined, but only a subset of the species has been counted.

Next, we use the species importance to build transfer function models with successively increasing number of species. We start with the two most important species and increase the number according to the species importance ranking. The performance of each model (i.e the prediction

error) was assessed using h-block cross validation to account for the effect of spatial autocorrelation in the training set (Telford and Birks, 2009). We used a cut off distance of 850 kilometres, which was shown to be appropriate for the North Atlantic (Trachsel and Telford, 2016). Because of the presence of cryptic species with specific ecological preferences, we have carried out the analyses separately for individual ocean basins following Kucera et al. (2005), assigning the new Red Sea samples of

ForCenS to the Indian Ocean. The discussion below focusses on the North Atlantic Ocean because of the abundance of both core top and down core data, but the species ranking for the other oceans is provided in the supplementary information. All calculations were carried out in R (R core team, 2016) using the packages rioja (Juggins, 2017), raster (Hijmans, 2017), vegan (Oksanen et al., 2018) and ggplot (Wickham, 2016).




**3 Results and discussion**

**3.1 Species importance ranking**

Irrespective of which regional training set is used, for both WA and MAT only a small number of
species appears important (Fig. 1, supplement). As shown by the example of the North Atlantic, cross
validation of the transfer function models with increasing number of species shows that there is a
large tail of low-importance species that do not add information and hence do not lead to improved
transfer function performance (Fig. 1). In general, it appears that reconstructions with similar errors
to the full species reconstruction can be achieved with less than a third or the total number of
species. To reach prediction errors at, or very close to, the possible minimum, fewer species are
needed for MAT than for WA. The species importance ranking varies somewhat by method and
region, but is generally similar (supplement). In the North Atlantic there is a seventy percent overlap
of the species in the top ten, with the most notable exception being the cold-water species *N.
pachyderma*, which appears unimportant for MAT, whereas it is ranked as number one for WA.
To understand why some species are more important than others, we consider their overall
abundance and the width of their thermal niche in the training set (Fig. 2). Species that appear
important for transfer function performance are in the first place those that are abundant (Fig. 2).
There are nevertheless differences between the two methods: for MAT inclusion of warm-water
species only appears sufficient to obtain a minimum prediction error, whereas for WA cold-water
species are also required. This is in line with 'global' nature of the technique, which requires
characterisation of thermal niches of the species across the entire temperature range of the training
set. The former likely reflects the implicit inclusion of information on the abundance of the remaining
(cold-water) species and analogues are found because the similarity is based on relative abundances
with respect to all species.
The analysis also reveals that there are many uninformative species, but very few – if any – real
nuisance species, i.e. species that lead to an increase in the prediction error when included in the
transfer function model. Possible species in this category are *G. glutinata* in the case of WA and *G.
bulloides* for MAT (Fig. 1). Both are species with a very wide thermal tolerance and a high degree of
cryptic diversity (Darling et al., 2017; André et al., 2014) suggesting that their response to
temperature may be complex.

**3.2 Effect on reconstructions**

Despite the apparent lack of importance of many species in the transfer function model
development, their inclusion matters for the reconstruction based on fossil assemblages. Transfer
function models with pruned species numbers yield reconstructions that are systematically different
from reconstructions with all species included (Fig. 3). The differences can be up to several degrees



centigrade and variable in time and space, with important implications for paleoceanographic
interpretations. As expected, the average difference between a reconstruction with all species
included and pruned species reconstructions, decreases with increasing number of species (Fig. 4).
Importantly, species pruning does not only result in more variability in the reconstructions, but leads

to a real bias towards either lower or higher temperatures than a reconstruction with all species (Fig.
4). Especially in the case of the MARGO reconstruction using MAT, the reconstruction with the
minimum number of species is fundamentally different from the one that uses all species (mean
absolute difference > 4 °C), even though the prediction errors are virtually identical. For WA the
average LGM cooling is relatively insensitive to species pruning (Fig. 4), yet, as for MAT, the spatial

pattern of reconstructed temperatures is, showing consistently lower temperature off northwest
Africa (Fig. 3B).

Inclusion of the most important species leads to rapid decrease in the difference between species-
pruned and all-species reconstructions, but unlike the species importance assessment (Fig. 1), adding
more species leads to further stepwise decreases in the difference (Fig. 4). Importantly, steps in

reconstruction difference occur with the addition of different species in different cores, indicating
that they result from specific downcore changes in the assemblage composition. The exception
appears to be *G. glutinata* in the case of WA, which leads to an increase both in the prediction error
and the difference of the reconstruction, supporting its potential rating as a nuisance species.

When adding species to the transfer function model, the reduction in the difference from the

reconstruction with all species is initially accompanied by a reduction in the prediction error (Fig. 5).
However, once the most important species are included, the prediction error stabilises whilst the
difference continues to decrease. This means that for the same fossil assemblage, multiple different
reconstructions with the same prediction error are possible. This pattern is visible in data sets from
different faunistic and climatic regions and different time spans. Moreover, it holds for both

methods, suggesting that it is a general pattern of the effect of species selection on SST
reconstructions using planktonic foraminifera assemblages. In WA species pruning has a greater
effect on the reconstruction error than on the differences between the reconstructions. The opposite
is visible for MAT, but for both methods species pruning leads to different reconstruction with the
same error. Taken at face value, and in the absence of independent evidence, such inherent

ambiguity renders it impossible to decide which of the reconstructions is more realistic. This adds a
previously unrecognised source of uncertainty to quantitative assemblage-based reconstructions.

### 3.3 Sensitivity to species pruning

To evaluate the cause of the sensitivity of reconstructions to species selection, we calculate for each

fossil sample a sensitivity measure based on the standard deviation of the reconstructions using



different number of species and evaluate its predictability by properties of the assemblages (Fig. 6).
The pruning sensitivity is all fossil datasets higher for MAT than for WA (Fig. 6). This partly reflects
the fact that WA requires more species to reach minimum prediction error (Fig. 1) and the range of
reconstructions used to calculate the pruning sensitivity includes fewer transfer function models with

more species included than for MAT.

Intuitively, it would be expected that the effect of species pruning will be larger in low-diversity
assemblages. However, for both techniques and all datasets, pruning sensitivity is unrelated to the
number of species present in the fossil sample (Fig. 6). The fact that pruning sensitivity is not related
to richness confirms that diverse assemblages contain many unimportant (redundant or

uninformative) species. Especially for MAT, it could be expected that fossil samples with a
composition that differs most from the training set will be most sensitive to pruning. This could be so
if some species that are judged as uninformative in the training set carry environmentally relevant
information downcore. Indeed, we observe such effect of analogue quality, with poorer analogue
assemblages being more sensitive to pruning (Fig. 6). As expected, the effect is stronger for MAT, but

only visible in the downcore records. However, the fact that the effect is generally weak and the
absence of a clear common pattern in species pruning sensitivity (Fig. 6) highlights the difficulty in
attributing assemblage changes to a single environmental factor and suggests that each site, or time
slice, has unique species turnover dynamics. The uniqueness of species dynamics in each dataset is
also reflected in the different position of the distinct steps in the difference between the

reconstructions with all species and the pruned reconstructions (Fig. 3). Because these steps are not
accompanied with a change in the transfer function performance, they indicate changes in the fossil
assemblages that either reflect a temperature sensitivity of the species that is not captured in the
training set (inadequate training set) or are not related to temperature (non-analogue condition
(Hutson, 1977)). Non-analogue situations could arise either secondarily from post-depositional

changes in the assemblages, or primarily and reflect a biological response to another environmental
variable than temperature, which is not apparent in the training set because of temporally variable
covariation between this predictor variable and temperature.

Post-depositional changes in species assemblages could arise from processes like dissolution (Berger,

1968), expatriation or sediment (and thus fossil) mixing due to bioturbation (Hutson, 1977; Kucera et
al., 2005). The effect of mixing should be most pronounced around intervals of assemblage change
and may create fossil species assemblages with poor analogues in the training set. We have assessed
to what degree analogue quality (dissimilarity from the training set) varies as a function of change in
the assemblages for the downcore records investigated here (Fig. 7). The observed relationship has

the right direction, but is weak and not apparent in each time series, suggesting that analogue quality



varies as a function of multiple parameters and does not simply reflect sediment mixing. In the analysed fossil samples, dissolution should be negligible and expatriation is unlikely to have changed considerably between the training set and fossil samples. We thus infer that the major reason for the observed ambiguity of the reconstructions is the effect of nuisance (non-temperature) variables on

fossil assemblage composition.

### 3.4 Implications for paleoecological reconstructions

Despite its intuitive simplicity and in spite of apparent statistical support (Morey et al., 2005), relating species assemblage change to a single environmental variable is not trivial (Juggins, 2013; Telford

and Birks, 2009; Telford and Birks, 2005; Telford et al., 2013). This means that quantitative reconstructions based on transfer functions must be interpreted with caution and evaluated in view of additional, independent evidence. Our analysis provides further evidence for the temporal emergence of apparently non-temperature related changes in fossil planktonic foraminifera assemblages that are not captured by calibration (Telford et al., 2013; Steinke et al., 2008; Siccha et

al., 2009). The ambiguity in reconstruction due to species selection thus highlights the potential of environmental variables that appear unimportant in shaping species communities today, in explaining changes in past assemblages. So, the important question is: how can this ambiguity be considered when interpreting individual reconstructions? We recommend as a first step to use the approach outlined here (ranking of species importance for each basin is provided in SI) to quantify

the sensitivity of reconstructions to pruning and evaluate its potential geological and biological drivers and use this as an indication of the influence of secondary/nuisance variables on the reconstruction. Although a method to translate the effect of pruning into a mechanistic and quantifiable uncertainty estimate is not available, one may conclude that reconstructions that are highly sensitive to species pruning may indicate that the observed assemblage changes cannot be

attributed solely to the environmental variable that is to be reconstructed.

Ultimately, we need a more mechanistic understanding of the factors that determine species assemblage composition in the sediment. Importantly, planktonic foraminifera species inhabit vertically and seasonally distinct habitats (Jonkers and Kučera, 2015; Rebotim et al., 2017). Thus, in almost all cases the species in a sediment assemblage have never actually lived together and their

abundance is thus unlikely to reflect the same forcing. Moreover, the controls on vertical and seasonal abundance variability are species specific, adding even more complexity to deriving a single environmental variable from an assemblage of different species. Ecological models that explicitly simulate assemblages from multiple environmental variables (Kretschmer et al., 2018; Lombard et al., 2011) may aid to improve our capabilities to quantitatively reconstruct past environmental



change from species assemblages. In a climate modelling context, such forward modelling of fossil
        assemblages is likely more fruitful than directly comparing the inferred temperatures.

## Conclusions

There are both theoretical and practical reasons to investigate whether full taxonomic resolution is
required to infer paleoenvironmental data from microfossil assemblages. We have addressed this
        issue using planktonic foraminifera, but we believe that our results are relevant to other groups of
        (marine) microfossils too. We have ranked species according to their importance for transfer
        function model development and shown that less than a third of the species is needed to derive a
        model that performs as good (or better than) a model with full taxonomic resolution. Nevertheless,
even though addition of more species has little effect on the transfer function model performance,
        sea surface temperature reconstructions with increasing number of species, are different. Thus,
        multiple different reconstructions are possible and their reliability cannot be assessed by transfer
        function performance in the training set. Our analysis suggests that fossil assemblages do not
        uniquely reflect a single environmental variable (sea surface temperature in this case), but rather
provide an integrated response to biotic (but not temperature related) and abiotic (sediment mixing)
        factors. We have identified a new way to detect uncertainty (ambiguity) in transfer-function
        reconstructions due to nuisance variables. The sensitivity of reconstructions to species selection can
        be quantified using the approach outlined here, facilitating an assessment of the robustness of the
        reconstruction.

## Acknowledgements

We thank Steve Juggins for advice at the initiation of this project. LJ acknowledges support from the
climate modelling initiative PalMod, funded by the German Federal Ministry of Education and
Research (BMBF).

## Data availability

All datasets used in this study are in the public domain: ForCens:
https://doi.pangaea.de/10.1594/PANGAEA.873570; WOA:
https://www.nodc.noaa.gov/OC5/WOA01/pr_woa01.html; Core MD95-2040:
https://doi.pangaea.de/10.1594/PANGAEA.66811; Core M35003-4:
        https://doi.pangaea.de/10.1594/PANGAEA.55756; MARGO LGM:
        https://doi.pangaea.de/10.1594/PANGAEA.227329. Code is available upon request from LJ.





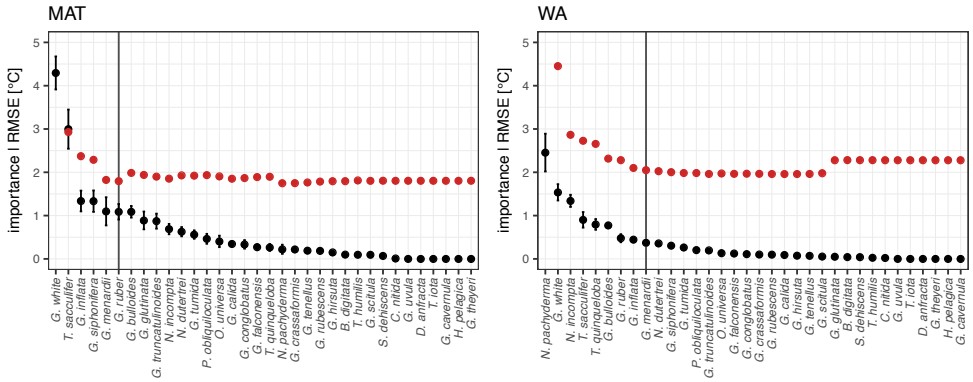


Figure 1: species importance (black) and transfer function performance (red) for models including all

species to the left of the point (e.g. the first red point for the MAT transfer function for the North

Atlantic denotes the prediction error (RMSE) of a model with *G. ruber* (white) and *T. sacculifer*). Error

bars on the species importance show the standard deviation of 10 replicates; see methods for

details. The dark vertical lines indicate the species needed for a transfer function model with

minimum error (MAT), or where including more species only leads to an incremental reduction in the

reconstruction error (WA).





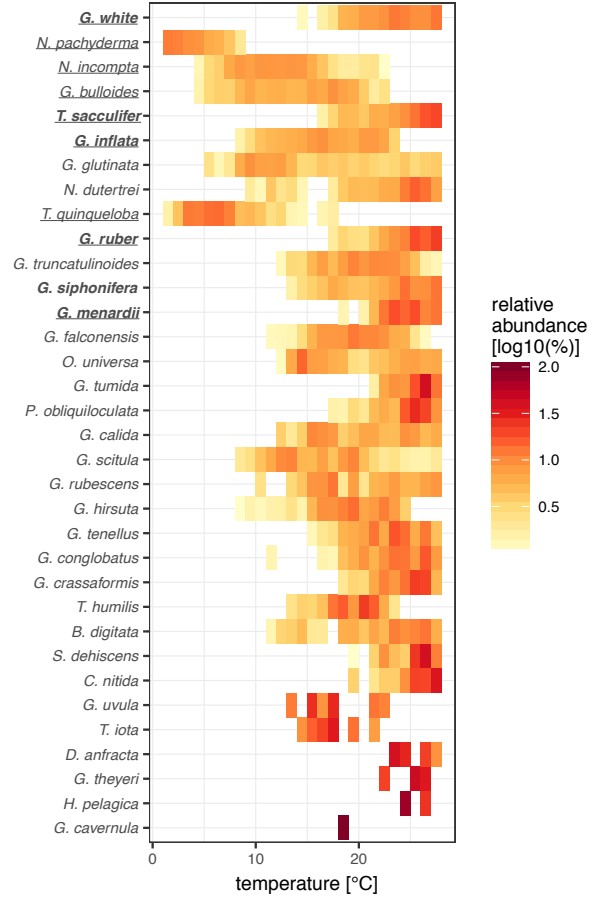


Figure 2: species thermal niche, ranked by average sedimentary abundance (top to bottom). Colours indicate average abundance (on a logarithmic scale) within one-degree centigrade bins. The essential species (i.e. needed for a transfer function model performance with minimum prediction error) for
MAT are marked in bold and those for WA are underlined. Despite their extremely well-defined thermal niche, rare species are not essential for transfer functions and those that are important are all abundant.



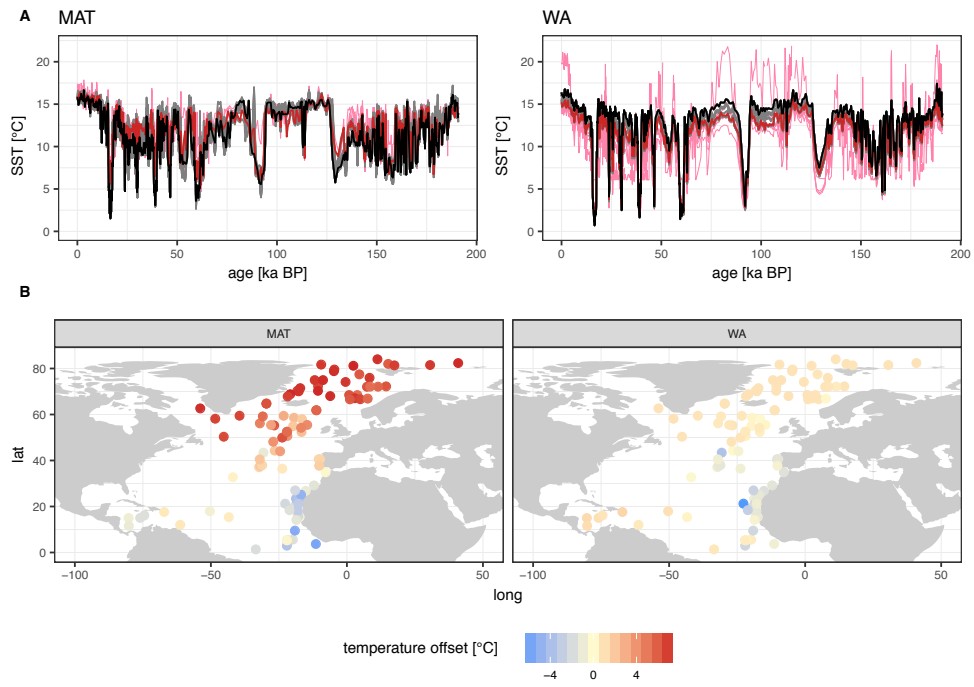


Figure 3: species selection changes SST reconstructions. A: SST reconstructions for core MD95-2040 from the Iberian Margin using 33 possible transfer function models with the number of species increasing according to their ranking. Thin pink lines are the reconstructions with fewer than the minimum number of species, red lines show the reconstructions with essential species only, dark

grey lines the reconstructions with more species and the black line is the 'final' reconstruction with all species. B: Spatial pattern of the difference between the SST reconstruction for the LGM with the minimum and with all species included.





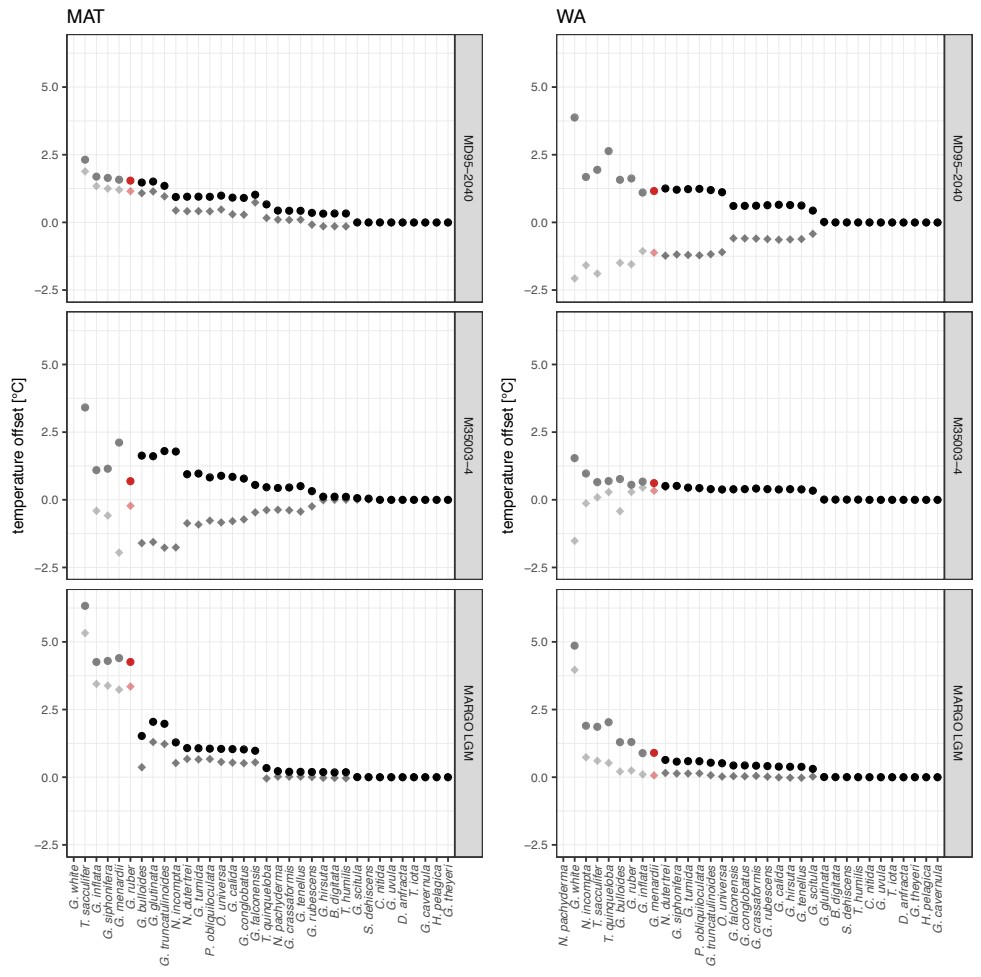


Figure 4: stepwise effect of species selection on SST reconstruction. Graphs show mean difference between reconstruction using transfer functions with the species up to the marked point (similar to Fig. 1). Grey symbols are the reconstructions with fewer than the minimum number of species and red symbols the reconstructions with the minimum number of species. Dots are the average of the mean absolute difference, diamonds the mean difference.




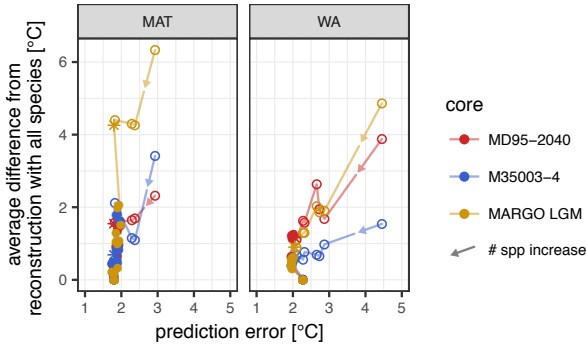

Figure 5: general effect of species pruning on reconstructions and on the prediction error. Each dot
represents a reconstruction with n species and its associated prediction error; arrows indicate
direction of increasing species numbers and decreasing species importance. Open symbols highlight
reconstructions with the essential species of which the inclusion leads to a reduction in the
prediction error. The reconstruction with minimum error and minimum number of species is marked
by a star. All other reconstructions are plotted as dots. Note that for MAT and WA the inclusion of
the essential species leads to a reduction of the prediction error, but that once these species are
included there are still multiple reconstructions with the same error possible.




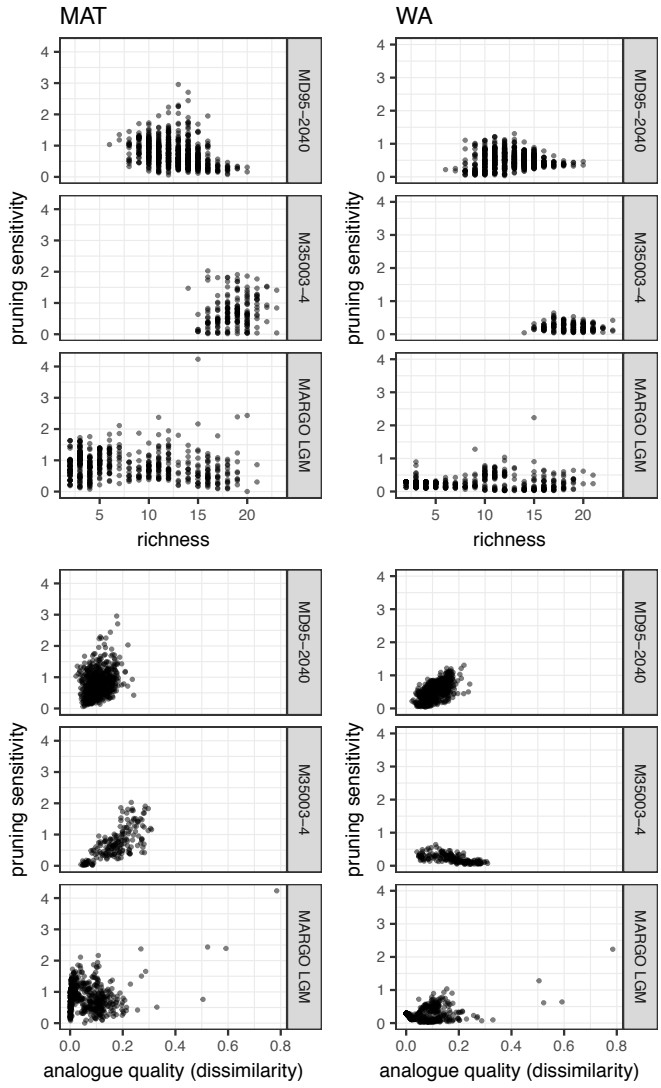

Figure 6: exploring the effect of species richness and analogue quality on the sensitivity to species

pruning. This sensitivity is defined as the standard deviation of the reconstructions with different

numbers of species, but similar prediction errors (i.e. those to the right of the red dot in Fig. 4).

Analogue quality is defined as the average dissimilarity from the five most similar samples in the

training set; zero values thus mean perfect analogues. A clear effect of richness cannot be observed,

yet for some of the downcore reconstructions (MD95-2040 and M35003-4) poor analogues appear

associated with higher pruning sensitivity, particularly for MAT.





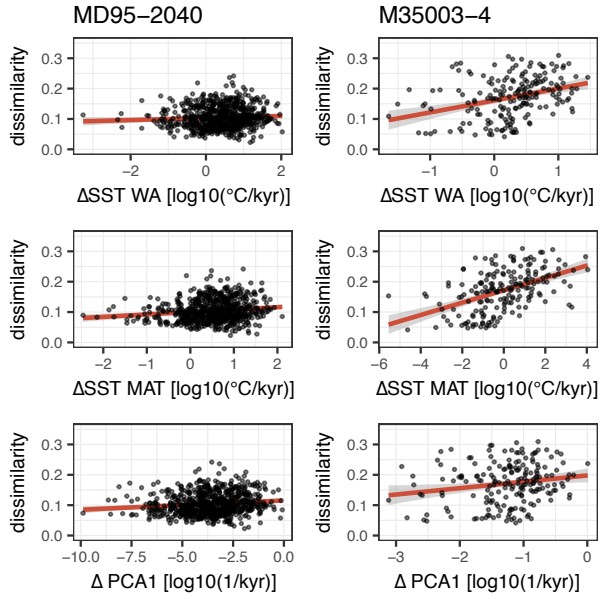

Figure 7: a possible role for sediment mixing in determining analogue quality (dissimilarity, see Fig. 6). Sediment mixing would create poor analogue assemblages when different species communities are mixed. The difference between subsequent species communities in the sediment core is here defined in two ways, i) as the difference in inferred temperature (top and middle row) and ii) as the first derivative of the loadings of the first axis of a PCA on the species abundances. Linear regression including 95 % confidence interval, is shown as a red line with grey shading. Only for core M35004-3 a weak positive relationship is visible, indicating that sediment mixing is a possible contributor to creating poor analogue assemblages, but cannot explain them entirely.




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
