# Peer review of "Sensitivity to species selection indicates the effect of nuisance variables on marine microfossil transfer functions"

_Climate of the Past, 2018_

## Referee Comment (RC1) · Anonymous Referee #1 · 16 Oct 2018

I have reviewed the manuscript presented by the authors and have found it to be well written, interesting and concise. The topic it addresses has indeed been genrerally overlooked by the wider community and there is no doubt that more work of this sort is necessary in the field of paleoclimatology.

They begin by ranking the species by importance. The method considered from Juggins et al. (2015) seems valid, but I wonder whether other methods could be considered and if the results would be significantly different; I imagine not so much. The authors then showed that for both MAT and WA, it is possible to obtain rather different paleo reconstructions when using a different number of species for the calibration of

the model, even though those models had very similar error of prediction in the training dataset. This underlines that the prediction error in the training dataset is not by itself a sufficient metric for characterizing the uncertainty in reconstructions. This also applies to other microfossils. It would be interesting to analyze the spatial patterns of the prediction errors in order to identify the sources of variation related to species-pruning.

The manuscript leaves many questions unanswered, but I understand that not all questions can be answered in a single article. I will be looking forward for future work concerning the quantification of the species-pruning uncertainty, and the identification of non-climatic biotic and abiotic factors leading to the uncertaitnties.

I found the treatment of the authors satisfying and did not find any important flaw with the analysis presented. Therefore, I recommend that it can be published without major revisions, and below I include a list of specific points which should be taken into account to improve the quality of the manuscript, particularly many figure captions should be revised.

Minor corrections: Line 15: 'information [about] all species'

Line 108: If they were not reported, I imagine they were not counted. I am not sure what the authors did, but for consistency, such species should then also be removed from the calibration dataset.

Lines 113: How many samples are kept for this part used to build the transfer function? I found the brief explanation hard to understand and had to read Juggins to understand. I would suggest rewriting this paragraph in a more straightforward way.

Line 118: I do not understand why the approach is repeated 10 times after the 1000 bootstrap. Couldn't the error estimate be obtained from the bootstrap estimates directly? Is it that the $\frac{1}{3}$ selection of species is the same for any given set of 1000 bootstraps?

Line 123: Does this mean you do not renormalize the species abundances after selecting a subset of species? If yes, then couldn't this lead to problems if for example we have two sites with a similar composition regarding the 'useful' species which are sensitive to say temperature, but one has a large amount of irrelevant species temperature-wise while the other has almost none.

Line 150: I would indicate the number of species in parenthesis for MAT (around 6 species) and for WA (around 9 species).

Line 201: Add comma after 'In WA'

Line 240: Is expatriation a synonym for sediment mixing or a different process? It is mentioned, but neither defined or discussed really.

Figure 1: Not sure that incremental is the right work, an incremental change could be large. Maybe "marginal" or a synonym would be more appropriate.

Figure 3: Could you define what years you are considering for the LGM. Even better would be to add a shaded background over that period which you averaged in the top row graphs.

Figure 4: The caption could be rewriten more clearly, it is not clearly stated that the difference is with the reconstruction with the full taxonomic resolution.

Figure 5: Is the prediction error calculated using the full timeseries or only the LGM as previously? Also, it might be useful to indicate the number of species used for the 1st point, does it start at n=2? I imagine that the increments of increasing number of species is simpy 1, i.e. ni+1=ni+1. Cool figure.

Figure 6: Second sentence could be revised grammatically speaking.
* * *

---

## Referee Comment (RC2) · Anonymous Referee #2 · 26 Nov 2018

This manuscript uses the methods developed by Juggins et al (2015) to determine the importance of each planktic foraminifera for the performance of transfer functions calibrated against sea-surface temperature. These results are used to make models with a greatly reduced set of species which perform almost as well as models based on the full set, however, the reconstructions were different.

The manuscript is generally well written.

I found some of the results surprising. In particular, the low importance of N. pachyderma with the modern analogue technique (MAT). This is an abundant (up to 100%) and common taxon with a clearly defined thermal niche. I found the explanation in

the manuscript reasonably persuasive for transfer function performance, but not for importance.

Intrigued, I re-implemented the method and, with a smaller training set, and found N. pachyderma to be among the most important taxon for MAT. Subsequently, I was given access to the authors' code which gave the same result. The authors blame a glitch and now get the same result. I encourage the authors to adopt the techniques described in the British Ecological Society guide to reproducible code.

At least some of the subsequent results will be affected by this glitch.

The manuscript gives some consideration as to why some taxa are not important, but it would be interesting to see this expanded. I can think of several reasons. Taxa with low importance might be nuisance taxa, these should have negative importance. Taxa with poorly defined niches, or broad tolerances will have low importance and may be good to exclude from transfer functions. In contrast, taxa with low abundances or occurring in only a small number of sites may have low importance as the analysis will give these low weight even though they might be valuable indicators. It would be useful to try to quantify how much these factors contribute to low importance.

Whereas importance can be evaluated objectively, the effect of species inclusion rules on reconstructions cannot be as the truth is not known. Adding more taxa to the reconstruction will obviously tend to make the reconstruction more similar to the all-taxa reconstruction. But any individual taxa could make the difference larger. Contrary to line 192, I do not regard the increase in difference with the inclusion of G glutinata as evidence that it is a nuisance taxon. It might be more powerful to run this analysis on the calibration set.

I am not sure we need to see the reconstructions with in 3A with fewer than the "minimum number" of taxa, as no one should be using these.

The bias observed in for MAT in figure 3B looks much like what I would expect given N.

pachyderma had been omitted.

It would be interesting (line 210) to compare the sensitivity of WA and MAT on the same range of models to identify the inherent variability of each.

Line 268. "that reconstructions that are highly sensitive to species pruning may indicate that the observed assemblage changes cannot be attributed solely to the environmental variable that is to be reconstructed"

This is an interesting idea, but I don't think this conclusion is justified given the results in the current version of the manuscript.

Whereas the manuscript demonstrates that only a few taxa are important for temperature reconstructions, I would hope that micropalaeontologists continue to count the full assemblage so that a range of questions can be addressed with the data. Only for routine analyses (for example water quality monitoring) is identifying only the important taxa justified.

Minor points

Line 176. Celsius not centigrade

Capitalisation of axis labels in all figures needs to be checked.

Figure 2. Might be better to use scale_fill_continuous(trans = "log", breaks = ...) than to log transform the percent.

The authors report (line 312) that the code is available on request. It would be much better to archive the code on, for example, github, or better still a permanent archive such as zenodo.org, ideally before review.

---

## Author Comment (AC1) · 15 Jan 2019

We would like to take this opportunity to thank the reviewer for their helpful comments on our manuscript. We will implement all suggested changes in a revised manuscript. Below we have copied the reviewer's comments and added our response in red and proposed changes to the manuscript in blue.

With kind regards,

Lukas Jonkers and Michal Kucera

============================================================================

I have reviewed the manuscript presented by the authors and have found it to be well written, interesting and concise. The topic it addresses has indeed been genrerally overlooked by the wider community and there is no doubt that more work of this sort is necessary in the field of paleoclimatology. They begin by ranking the species by importance. The method considered from Juggins et al. (2015) seems valid, but I wonder whether other methods could be considered and if the results would be significantly different; I imagine not so much. The authors then showed that for both MAT and WA, it is possible to obtain rather different paleo reconstructions when using a different number of species for the calibration of the model, even though those models had very similar error of prediction in the training dataset. This underlines that the prediction error in the training dataset is not by itself a sufficient metric for characterizing the uncertainty in reconstructions. This also applies to other microfossils. It would be interesting to analyze the spatial patterns of the prediction errors in order to identify the sources of variation related to species-pruning.

The manuscript leaves many questions unanswered, but I understand that not all questions can be answered in a single article. I will be looking forward for future work concerning the quantification of the species-pruning uncertainty, and the identification of non-climatic biotic and abiotic factors leading to the uncertaitnties.

I found the treatment of the authors satisfying and did not find any important flaw with the analysis presented. Therefore, I recommend that it can be published without major revisions, and below I include a list of specific points which should be taken into account to improve the quality of the manuscript, particularly many figure captions should be revised.

We thank the reviewer for the encouragement. We agree with the evaluation that some issues are left unanswered. One of these – the ecological reasons for species ranking – was also highlighted by reviewer 2, and we thus decided to expand the discussion on this particular topic. In addition, we would like to highlight at this point that due to an error discovered by reviewer 2, the species importance ranking for MAT in the North Atlantic has changed. This change only applies to the importance ranking of the species, it has no fundamental implications for the conclusions of our manuscript, but it leads to substantial changes to the discussion. For more details, please refer to our response to reviewer 2.

Minor corrections:

Line 15: 'information [about] all species'

Done.

Line 108: If they were not reported, I imagine they were not counted. I am not sure what the authors did, but for consistency, such species should then also be removed from the calibration dataset.

We agree with the reviewer that this is potentially an important point. However, when counting foraminifera assemblages for SST reconstructions it is common practice to count all species and we assume here that species that were absent in the entire dataset (e.g. tropical species in a polar assemblage) were simply not reported (or, conversely, the authors only reported species which occurred at least once in their dataset). Such treatment of missing species is a common practice with synthesis datasets in paleoecology (e.g. MARGO).

Lines 113: How many samples are kept for this part used to build the transfer function? I found the brief explanation hard to understand and had to read Juggins to understand. I would suggest rewriting this paragraph in a more straightforward way.

Sample selection for the bootstrap and OOB is carried out randomly with replacement. This means that approximately 63 % of the samples are used to build the model and the remainder is used as the OOB. We will add this to the description of the method.

Line 118: I do not understand why the approach is repeated 10 times after the 1000 bootstrap. Couldn't the error estimate be obtained from the bootstrap estimates directly? Is it that the 1/3 selection of species is the same for any given set of 1000 bootstraps?

We realised that a very large number of bootstrap samples is required the obtain stable results. One could of course also obtain the error directly from the bootstrap estimates, but the results would be similar and in this way we could make use of R packages without too much modification of the code. The species selection is variable and random across all bootstraps.

Line 123: Does this mean you do not renormalize the species abundances after selecting a subset of species? If yes, then couldn't this lead to problems if for example we have two sites with a similar composition regarding the 'useful' species which are sensitive to say temperature, but one has a large amount of irrelevant species temperature-wise while the other has almost none.

The reviewer is right in that there is no re-normalisation; all remaining species are included in (a virtual) rest group. In this way, the potential issue, quite correctly raised by the reviewer, is circumvented. This procedure also mimics a common situation where researchers who only counted the most abundant species also often provide the total number of planktonic foraminifera.

Line 150: I would indicate the number of species in parenthesis for MAT (around 6 species) and for WA (around 9 species).

Will do.

Line 201: Add comma after 'In WA'

We will reword the sentence.

Line 240: Is expatriation a synonym for sediment mixing or a different process? It is mentioned, but neither defined or discussed really.

Expatriation refers to the incidental/occasional advection by ocean currents of species outside their normal habitat. We realise that this is in fact not a post-depositional process and will omit it here.

Figure 1: Not sure that incremental is the right work, an incremental change could be large. Maybe "marginal" or a synonym would be more appropriate.

We agree and will follow the suggested rewording.

Figure 3: Could you define what years you are considering for the LGM. Even better would be to add a shaded background over that period which you averaged in the top row graphs.

We follow the MARGO definition of LGM (23 to 19 kyr BP), but this is strictly speaking irrelevant here. The different figures are showing two entirely independent datasets. We will highlight this in the caption. The two datasets are used to show the influence of species selection in two different situations (time-slice and time-series).

Figure 4: The caption could be rewriten more clearly, it is not clearly stated that the difference is with the reconstruction with the full taxonomic resolution.

We apologise, some text appeared to have gone missing. We will change the caption to: '*Figure 4: effect of species selection on SST reconstruction. Graphs show mean difference between reconstructions using species-pruned transfer function models and the model that includes all species. The results are ordered according to species importance, with each dot representing the result from a transfer function model with the species up to the marked point (similar to Fig. 1). Grey symbols are the reconstructions with fewer than the minimum number of species and red symbols the reconstructions with the minimum number of species. Dots are the average of the mean absolute difference, diamonds the mean difference.*'

Figure 5: Is the prediction error calculated using the full timeseries or only the LGM as previously? Also, it might be useful to indicate the number of species used for the 1st point, does it start at n=2? I imagine that the increments of increasing number of species is simpy 1, i.e. ni+1=ni+1. Cool figure.

The reviewer is correct and the start is at two species and the increments are always a single species. We will add this to the figure caption. However, the prediction error is based on the cross-validation of the transfer function model in the core top data set and thus independent of the reconstruction.

Figure 6: Second sentence could be revised grammatically speaking.

For clarity we will change this sentence to: '*This sensitivity is defined as the standard deviation of the reconstructions with more than the minimum required number of species (i.e. based on transfer function models with uninformative species, those to the right of the red dot in Fig. 4).*'

---

## Author Comment (AC2) · 15 Jan 2019

We would like to thank the reviewer for their critical look at our manuscript and are grateful for spotting an unexplained mistake in our analysis (species ranking in the North Atlantic). We have repeated all analysis and will make the code publicly available. Whilst the species ranking for MAT in the North Atlantic is indeed different from our original submission, the conclusions of our study remain unaffected: the sensitivity to species pruning provides a useful means to assess the influence of other environmental variables in transfer function-based reconstructions. However, the now correct (and replicable) species ranking leads to changes in the discussion. Below we have copied the reviewer's comments and provided our response in red and proposed changes are indicated in blue.

With kind regards,

Lukas Jonkers and Michal Kucera

================================================================================

This manuscript uses the methods developed by Juggins et al (2015) to determine the importance of each planktic foraminifera for the performance of transfer functions calibrated against sea-surface temperature. These results are used to make models with a greatly reduced set of species which perform almost as well as models based on the full set, however, the reconstructions were different.
The manuscript is generally well written.

I found some of the results surprising. In particular, the low importance of N. pachyderma with the modern analogue technique (MAT). This is an abundant (up to 100%) and common taxon with a clearly defined thermal niche. I found the explanation in the manuscript reasonably persuasive for transfer function performance, but not for importance.

Intrigued, I re-implemented the method and, with a smaller training set, and found N. pachyderma to be among the most important taxon for MAT. Subsequently, I was given access to the authors' code which gave the same result. The authors blame a glitch and now get the same result. I encourage the authors to adopt the techniques described in the British Ecological Society guide to reproducible code.

At least some of the subsequent results will be affected by this glitch.

Repeat analysis showed indeed that the species importance ranking for MAT in the North Atlantic was erroneous. Our new analysis shows indeed that N. pachyderma is the most important species (see figure below).

[Figure]

This ranking has some consequences for the remaining part of the manuscript:

i)    it affects the number of species required to achieve a prediction error similar to the solution with all species;

ii)   the species importance ranking is now more similar for both methods;

iii)  it affects to some degree the reconstructed temperatures (Figs 3 and 4).

However, the most important finding of our analysis remains unaffected: we still find many different reconstructions with the same prediction error (see figure below)

[Figure]

Based on the prediction error alone, these reconstructions appear equally valid, thus highlighting a previously unrecognised source of uncertainty in transfer function based environmental reconstructions. We will update the manuscript to reflect these changes.

Despite extensive efforts, we were not able to reconstruct where the error occurred. It probably resulted from using erroneously an earlier version of the analysis, or an issue with file naming. However, the error only affected the one analysis and after sharing the code between us and the reviewer, we could establish that the new ranking is replicable. We apologise for this mistake and applaud the reviewer for their stringency.

The manuscript gives some consideration as to why some taxa are not important, but it would be interesting to see this expanded. I can think of several reasons. Taxa with low importance might be nuisance taxa, these should have negative importance. Taxa with poorly defined niches, or broad tolerances will have low importance and may be good to exclude from transfer functions. In contrast,

taxa with low abundances or occurring in only a small number of sites may have low importance as the analysis will give these low weight even though they might be valuable indicators. It would be useful to try to quantify how much these factors contribute to low importance.

The reviewer points out an interesting aspect of our study and given the comments by reviewer 1 we expand the discussion on this topic. We agree with the reviewers' categorisation, although we would like to point out that we did not find any strict nuisance species with a negative importance. We assume that a species' importance for transfer function is dependent on i) its abundance, ii) its niche width and iii) its sensitivity to the environmental variable that best predicts the entire assemblage. To assess how these factors influence species importance we calculated average and maximum (99th percentile) abundance (in %), the thermal niche width (in deg C) and the temperature sensitivity (expressed as the goodness of fit (r2) of the species abundance to a Gaussian curve) for each species and assessed how well these parameters explained species importance using linear regression with increasing number of variables. We acknowledge that some of these variables are autocorrelated (e.g. temperature sensitivity and abundance are positively correlated, suggesting that the distribution of rare species is not primarily governed by temperature (in agreement with their low importance), or that our metric of temperature sensitivity is inadequate), but we believe this is of minor importance for the exploratory analysis we carry out.

For all ocean regions and for both MAT and WA we find that abundance appears to explain most of the variance in species importance. For the North Atlantic for instance the r2 of a linear model that only includes maximum abundance is 0.79 and 0.91 for MAT and WA respectively. Addition of niche width raises the r2 slightly to 0.83 and 0.93 and further addition of temperature sensitivity has a negligible effect (r2 0.84 and 0.94). This indicates that species abundance is the most important factor controlling species importance. This is probably due to the simple fact that abundant species are consistently present and allow best definition of thermal niche. We agree with the reviewer that rare taxa with a well-defined narrow niche might be good indicator species, but (if they exist) their usefulness for transfer functions is limited by their low abundance, leading to inconsistence incidence. We will add the following to the discussion:

We will add this discussion to the section where we consider species importance:

*To understand why some species are more important than others, we consider their overall maximum abundance, the width of their thermal niche in the training set and their temperature sensitivity as potential predictors of importance. We define temperature sensitivity based on how well the species abundance in temperature space can be described by a simple Gaussian curve. This analysis reveals that abundance (Fig. 2) is the best predictor of species importance (Table 1). Indeed, multiple regression models that include all three variables perform only marginally better than a*

*model using abundance alone. However, we note that all three variables are correlated to some degree. Interestingly, abundance and temperature sensitivity are positively correlated (r = 0.84), implying that the thermal niche of abundant species is better defined compared to rare species (Fig. 3A). We also observe that temperature sensitivity and thermal niche width are correlated. Counterintuitively, this correlation is positive: species with a narrow thermal niche appear less temperature-sensitive (r = 0.60; Fig. 3B). We attribute this pattern to a combination of low abundance of species with narrow thermal niches (sensitivity being correlated with abundance) and the possibility that their distribution is not primarily governed by temperature, assuming that the narrow thermal niche may be an artefact of adaptation to specific oceanic regions or regimes, only secondarily correlated with temperature.*

Whereas importance can be evaluated objectively, the effect of species inclusion rules on reconstructions cannot be as the truth is not known. Adding more taxa to the re- construction will obviously tend to make the reconstruction more similar to the all-taxa reconstruction. But any individual taxa could make the difference larger. Contrary to line 192, I do not regard the increase in difference with the inclusion of G glutinata as evidence that it is a nuisance taxon. It might be more powerful to run this analysis on the calibration set.

We agree with the reviewer as was reflected by our statement "*The analysis also reveals that there are many uninformative species, but very few – if any – real nuisance species…*". The reason why we assigned G. glutinata a possible nuisance status is for two reasons i) inclusion of this species in the calibration dataset leads to an increase in the prediction error (Fig. 1) and ii) inclusion of this species in the reconstructions leads to a jump in the difference in reconstructed temperature (Fig. 4). Our original submission already included this analysis, as reflected in the sentence "*G. glutinata in the case of WA, which leads to an increase both in the prediction error and the difference of the reconstruction, supporting its potential rating as a nuisance species*".

I am not sure we need to see the reconstructions with in 3A with fewer than the "minimum number" of taxa, as no one should be using these.

We agree. These will be removed from the figure.

The bias observed in for MAT in figure 3B looks much like what I would expect given N. pachyderma had been omitted.

This figure (now figure 4) will be changed to reflect the revised species importance ranking. The patterns for MAT and WA are now more similar:

[Figure]

It would be interesting (line 210) to compare the sensitivity of WA and MAT on the same range of models to identify the inherent variability of each.

The revised species importance ranking yields a top ten of most important species that overlap by 90 % (see figure above). The sensitivity of the results to the species importance ranking (the inherent variability) is therefore now (using the new ranking for NA) to a large extent assessed. We also note that the uncertainty in importance of the individual species are rather large, rendering the ranking of the unimportant species more uncertain. We see therefore little merit in repeating the analysis with 'swapped' species rankings.

Line 268. "that reconstructions that are highly sensitive to species pruning may indicate that the observed assemblage changes cannot be attributed solely to the environmental variable that is to be reconstructed"

This is an interesting idea, but I don't think this conclusion is justified given the results in the current version of the manuscript.

Our updated results still reveal the same patterns in the reconstructions and we find it hard to identify an alternative explanation, so we prefer to uphold the statement. We hope the revised analysis in the discussion will help to convince the reader that our conclusions are justified.

Whereas the manuscript demonstrates that only a few taxa are important for temperature reconstructions, I would hope that micropalaeontologists continue to count the full assemblage so that a range of questions can be addressed with the data. Only for routine analyses (for example water quality monitoring) is identifying only the important taxa justified.

We agree with the reviewer. Our intention was not to encourage counting fewer species, but to quantitatively investigate the effect of species pruning. This was motivated in part because some legacy datasets include only a limited number of species and we wanted to assess if these could be used. We will include an explicit statement in the revised version to encourage scientists to continue working with complete taxonomic resolution.

Minor points

Line 176. Celsius not centigrade

Addressed.

Capitalisation of axis labels in all figures needs to be checked.

Will do.

Figure 2. Might be better to use scale_fill_continuous(trans = "log", breaks = ...) than to log transform the percent.

OK.

The authors report (line 312) that the code is available on request. It would be much better to archive the code on, for example, github, or better still a permanent archive such as zenodo.org, ideally before review.

We agree with the reviewer. Our code is now available on:

https://github.com/lukasjonkers/species_selection and will be made available on zenodo upon acceptance of our manuscript.

---

## Author Response (AR2)

Dear Christian,

Please find our revised manuscript 'Sensitivity to species selection indicates the effect of nuisance variables on marine microfossil transfer functions'. First, we would like to thank the reviewer for their comments, which we have copied below. Our response is included in red font. Line numbers refer to the manuscript with tacked changes that is appended below.

We hope that our manuscript now merits publication in Climate of the Past.

With kind regards,

Lukas Jonkers and Michal Kucera

This authors have dealt with the problems in the first version of the manuscript and I now have only a few comments.

Line 45 "taxonomic resolution" I am not sure this is the best term to use. I see taxonomic resolution as recording the counts at the genera/species/subspecies level, whereas this ms discusses whether certain taxa can be omitted. Even higher taxonomic resolution, to cryptic species level, might solve much of the problem with e.g. G. glutinata. I don't have a better term, except perhaps "species coverage".
The reviewer is right. We have changed the wording to 'However, there are fundamental, theoretical reasons to question if inclusion of all species is necessary for accurate reconstructions.'. L 49-50.

Line 139 "ggplot2" not "ggplot"
Changed.

Line 156. The comparison of each species distribution to a Gaussian curve implicitly assumes that species are expected to have a Gaussian niche. While some theory suggests that they should, much field evidence shows that they often do not. This is important as WA assumes that the species niches are ~Gaussian. Species with strongly non-Gaussian niches may therefore appear to be unimportant or nuisance taxa. I think this is what is happening with G. glutinata. It has a strongly non-Gaussian niche with respect to temperature (which may of course be modulated by adjustment to the bloom season with latitude), being absent from cold water, most abundant in temperate water, and present at moderate abundances even in tropical waters. I strongly suspect that this non-Gaussian niche, rather than the importance of secondary variables, is what is making G. glutinata appear to be a nuisance taxon. The

interpretation of the importance statistics is a little more subtle than previously realised. G. glutinata is probably the most extreme taxa in this regard.

We agree and have added that the Gaussian niche shape is an assumption (L 161-162). As for *G. glutinata*, this species has indeed a thermal niche that cannot easily be described by a Gaussian curve. This may, as the reviewer suggests, affect its importance for WA transfer functions. However, whether this non-Gaussian niche reflects a complex response to temperature, or the influence of other variables is an open question.

Line 160. The estimation of the temperature sensitivity may be sub-optimal in that it does not account for the binomial nature of the data. A better (and easier) way to fit the estimated Gaussian curves would be with a binomial GLM using a quadratic model. The model pseudo-r2 can then be used.

We thank the reviewer for their suggestion and have tested, whether the suggested approach yields results different from our original. We find that the correlation patterns are unaffected, only the value of the correlation coefficient is affected. The correlation between niche width and temperature sensitivity decreases, but it does not affect our statement: '*We attribute this pattern to a combination of low abundance of species with narrow thermal niches (sensitivity being correlated with abundance) and the possibility that their distribution is not primarily governed by temperature, assuming that the narrow thermal niche may be an artefact of adaptation to specific oceanic regions or regimes, only secondarily correlated with temperature.*' We therefore prefer to keep the text as it is, in particular as this analysis just serves to gain some more insight into what may explain species importance. After all, the main goal of our manuscript is not to determine how to best describe a species thermal niche, but to investigate the influence of species selection on temperature reconstructions.

Perhaps more importantly, the reported correlation between abundance and temperature sensitivity may simply be an artefact of rare taxa being more noisy rather than any indication of their sensitivity to non-temperature gradients. It might be possible to test this with some appropriate null models.

We agree with the reviewer that the data on rare species are noisier and highlight this in the text "*We also note that the detection of rare species is prone to large uncertainty due to the limited number of specimens counted and thus the characterisation of their ecological niche is inherently more difficult*". Still, we would expect rare species to be present at more sites if their distribution was governed by temperature only. Since they are not, we therefore prefer to keep the remaining text as it is and consider the suggestion as an avenue for further research.

Line 170 "for MAT inclusion of warm-water species only appears sufficient to obtain a minimum prediction error, ..." Is this a relic from the first version? Both MAT and WA now report that the cold-water species N. pachyderma is important.

We apologise, this is indeed text from the previous version that should have been deleted.

Line 190 "celsius" -> "Celsius"

Corrected.

Line 265 "proof" is far to strong a term here. "evidence" perhaps. I am fairly certain that many of the same patterns of importance and changing reconstructions would be observed in artificial data where all taxa are sensitive to temperature only. As it stands, I am not totally convinced that the variability in reconstructions demonstrates the importance of secondary gradients.

Changed to evidence

Fig 4b improve or delete axis labels

Done

Table 1. "Correlation coefficients of linear regression" - is this the model R2 - the coefficient of determination?

We have corrected this.

line 275 This enjoinder to count all taxa (which I concur with) contradicts the suggestion in the introduction that a subset of taxa could be counted for speed.

We agree that there is some contradiction, but counting fewer species would speed up the analyses in principle. This was part of the motivation for carrying out this study, but we discovered that the implications of species pruning were such that we cannot recommend counting fewer species. We therefore would like to leave the text as it is, but we reformulated the sentence to make our recommendation clearer.

[revised manuscript text omitted]